# Metabonomics and Transcriptomics Analyses Reveal the Underlying HPA-Axis-Related Mechanisms of Lethality in *Larimichthys polyactis* Exposed to Underwater Noise Pollution

**DOI:** 10.3390/ijms252312610

**Published:** 2024-11-24

**Authors:** Qinghua Jiang, Yu Zhang, Ting Ye, Xiao Liang, Bao Lou

**Affiliations:** Zhejiang Key Laboratory of Coastal Biological Germplasm Resources Conservation and Utilization, Institute of Hydrobiology, Zhejiang Academy of Agricultural Sciences, Hangzhou 310000, China; jqh881130@163.com (Q.J.); zhangy@zaas.ac.cn (Y.Z.); 15tye@stu.edu.cn (T.Y.); liangxiao1225@yeah.net (X.L.)

**Keywords:** *Larimichthys polyactis*, underwater noise, HPA axis, metabolic disorders, metabonome, transcriptome

## Abstract

The problem of marine noise pollution has a long history. Strong noise (>120 dB re 1 µPa) will affects the growth, development, physiological responses, and behaviors of fish, and also can induce the stress response, posing a mortal threat. Although many studies have reported that underwater noise may affect the survival of fish by disturbing their nervous system and endocrine system, the underlying causes of death due to noise stimulation remain unknown. Therefore, in this study, we used the underwater noise stress models to conduct underwater strong noise (50–125 dB re 1 µPa, 10–22,000 Hz) stress experiments on small yellow croaker for 10 min (short-term noise stress) and 6 days (long-term noise stress). A total of 150 fishes (body weight: 40–60 g; body length: 12–14 cm) were used in this study. Omics (metabolomics and transcriptomics) studies and quantitative analyses of important genes (HPA (hypothalamic–pituitary–adrenal)-axis functional genes) were performed to reveal genetic and metabolic changes in the important tissues associated with the HPA axis (brain, heart, and adrenal gland). Finally, we found that the strong noise pollution can significantly interfere with the expression of HPA-axis functional genes (including corticotropin releasing hormone (CRH), corticotropin releasing hormone receptor 2 (CRHR2), and arginine vasotocin (AVT)), and long-term stimulation can further induce metabolic disorders of the functional tissues (brain, heart, and adrenal gland), posing a lethal threat. Meanwhile, we also found that there were two kinds of death processes, direct death and chronic death, and both were closely related to the duration of stimulation and the regulation of the HPA axis.

## 1. Introduction

The problem of marine noise pollution has a long history, and most of the causes are related to human factors, including offshore energy platforms, shipping, and scientific exploration activities [1,2]. In recent years, the noises produced by large ships or small boats have become the most ubiquitous and pervasive source of anthropogenic noise in the oceans [3,4,5]. Some research reports suggested that the noise caused by ships includes both high- and low-frequency noise, and the SPL (sound pressure level) of the main noise source has been found to range from 90 to 190 dB (re 1 µPa, 25–35,000 Hz) [5,6,7]. Underwater noise pollution seriously interferes with the unique auditory perception and sound communication systems of marine animals and poses a serious threat to their survival [8,9,10,11,12]. Strong underwater noise (>120 dB re 1µPa) affects the growth, development, physiological responses, and behaviors of marine animals [13,14,15,16,17,18,19]. It can also cause damage to their auditory organs, visceral tissues, and central nervous systems and disturb their hormonal endocrine regulation [20,21,22]. For instance, studies have shown that underwater noise stimulation can significantly increase the levels of adrenal steroid, arginine vasotocin (AVT, arginine vasopressin (AVP) homologues), glucocorticoids (GCs), catecholamines, aldosterone, and other hormones in the plasma of many aquatic mammals [23,24,25]. However, these studies did not provide a detailed explanation of the underlying causes of fish mortality due to underwater noise stimulation. This may be because these studies were conducted in uncontrolled environments; moreover, fish will flee immediately after perceiving a noise stimulus, and different fish have different adaptations to noise.

Many researchers have suggested that the abnormal responses of fish to underwater noise stimulation may be a physiological stress response [26,27]. The main regulatory axes that produce feedback to stress stimuli are the HPA (hypothalamic–pituitary–adrenal) axis and the SNS (sympathetic nervous system) [28,29], both important components of the central nervous system (CNS). Recent studies confirmed that the CNS is the main feedback control center determining reactions to underwater noise stimulation in aquatic animals [30]. In these reports, the researchers suggested that the HPA axis is the more critical and important component for the regulation of internal environmental homeostasis, especially during chronic stimulation, and its disorder can cause many adverse consequences for the body’s health, such as the deposition of visceral fat, cardiovascular disease, and autoimmune disease [28,29,30]. Most of these diseases are closely related to the heart, which may also indicate that underwater noise stress poses a threat to the functional stability of the heart. On the other hand, some reports have found that fish can adapt to noise over a period of time [31,32]. This may be related to their special ear structure; the sensory hair cells of a fish’s inner ears have some ability to recover from damage [33]. However, based on the research on stress adaptability, and the fact that most underwater noise pollution is chronic and persistent, the reason why fish adapt to noise stimulation may be more complicated than simply the recovery of hearing organ damage. It could also be that the HPA axis regulates and maintains the internal environmental homeostasis, disturbances of which may be the real cause of fish mortality due to underwater noise stimulation.

Recently, we found a fish species that could be a good subject for experiments on underwater noise-induced lethality. The small yellow croaker (*Larimichthys polyactis*) is a recently and successfully domesticated economic fish that is widely distributed in the Bohai Sea, East Sea, and Yellow Sea of China [34]. *Larimichthys polyactis* has auditory perception organs with otolith structures and is very sensitive to underwater environmental noise. *L. polyactis* has a complete cranial nervous system, including telencephalon, midbrain, cerebellum, and the medulla, allowing a rapid response to noise. When they are exposed to underwater noise (120 dB re 1 µPa) for a long time, they exhibit obvious avoidance behavior, and in severe cases, they lose the ability to maintain balance and become suspended in water in a state resembling thanatosis [35]. Thus, *L. polyactis* is very suitable for research into the damaging effects of noise stress on the brain and nerve system of fish.

In our study, to avoid uncontrollable environmental factors, we constructed an underwater noise stress model (Figure 1) (details are provided in the Materials and Methods Section). We used an underwater noise detection system to collect the engine noise of a common small fishing boat (original audio: 50–125 dB re 1 µPa, 10–22,000 Hz) to use as the underwater noise stimulus sound source. Finally, we conducted long-term (treatment for 6 days) and short-term (treatment for 10 min) underwater noise stress experiments in *L. polyactis*. We collected three important tissues associated with the HPA axis (brain, heart, and adrenal gland) from the surviving fish for transcriptomic and metabonomic sequencing analysis to measure the changes in gene expression and metabolite levels. Through these analyses, we investigated and evaluated the regulatory function of the HPA axis during survival, especially the changes in differential expression of its characteristic genes (corticotropin releasing hormone (CRH), AVT, pro-opiomelanocortin (POMC), etc.). In summary, this study attempted to analyze the changes in transcriptional and metabolic levels in the internal environment of fish under noise stress, revealed the influence of noise stress on the regulatory function of the HPA axis, and explored the correlation between the HPA axis and homeostasis, so as to supplement and explain the fundamental causes of death due to underwater noise stimulation.

## 2. Results

### 2.1. The Results of Noise Stress Experiments and Background Noise Detection

The SPL detection results of all environment noises are shown in Figure 2. The SPL of the indoor environment in the culture house over 24 h was 29–41 dB (re 20 µPa) (Figure 2A), and the underwater background noise in the control group over 24 h was 45–79 dB (1 µPa, 10–25,000 Hz) (Figure 2B). The SPL of the original audio of engine noise was 50–125 dB (re 1 µPa, 10–22,000 Hz), and in the noise stress experiment group, the real-time noise SPL over 24 h was 39–120 dB (re 1 µPa, 10–22,000 Hz) (Figure 2C). The results of the dead fish statistics are shown in Table 1; the death rate of the long-term noise stimulation group was 26.67% (8/30), and no deaths occurred in the short-term noise stimulation group. The brains of some dead fish in the long-term noise stimulation group showed obvious hemorrhagic symptoms (Figure 3).

### 2.2. Analysis of Transcriptomic Data

A total of 81.44 Gb of clean data were obtained from eighteen libraries, and every library contained at least 6.31 Gb of clean data. The Q30 value was over 93.45% (Appendix A). Additionally, a total of 63,928 assembled transcripts were obtained by mapping to the *L. polyactis* reference genome [36]. After database comparison, 30,032 genes were identified (20,920 known genes and 9112 new genes). All these raw sequence data were deposited in the National Center for Biotechnology Information (NCBI) Sequence Read Archive (SRA) (see the “Data Availability Statement” for details).

The results of the correlation analysis between biological replicate samples are shown in Figure 4A. All tissue samples were from the long-term noise stress experiment groups. Among them, the tissue samples of the experiment group fish included the brain group (E-Br-1, 2, and 3), the heart group (E-Ht-1, 2, and 3), and the adrenal gland group (E-AG-1, 2, and 3); meanwhile, the same tissue samples were collected from the control group (brain: C-Br-1, 2, and 3; heart: C-Ht-1, 2, and 3; adrenal gland: C-AG-1, 2, and 3). A total of 125 DEGs (differentially expressed genes) between the three groups were obtained (Appendix A), and the statistical comparison results are shown in Figure 4B. A KEGG (Kyoto Encyclopedia of Genes and Genomes) enrichment analysis of the three alignment groups was performed, and we removed the enrichment result of a single gene in the TR_Ht_vs_Control_Ht group (PLA2G4 (cytosolic phospholipase A2, also called CPLA2), evm.TU.Scaffold282.7, up-regulated) (Appendix A); the final enrichment analysis result is shown in Figure 5. The main enrichment pathways of the TR_Br_vs_Control_Br group (Figure 5A) included purine metabolism, hematopoietic cell lineage, and the Hippo signaling pathway. In the TR_AG_vs_Control_AG group (Figure 5B), the main enrichment pathways were the Hippo signaling pathway, thyroid hormone synthesis, autoimmune thyroid disease, etc. The results of the GO (Gene Ontology) functional enrichment-oriented analysis are shown in Figure 6. The red marked parts represent the functions in which DEGs were significantly enriched, and the arrow lines indicate the potential relationships between every function. The statistical results of expression differences in genes related to HPA axis function in each group are shown in Table 2. These functional genes included CRH, CRHR2, CRHBP (corticotropin-releasing factor binding protein), AVT, AVPR2 (vasopressin V2 receptor), POMC, MC2R (melanocortin type 2 receptor, also called ACTHR (adrenocorticotropic hormone receptor)), MC5R (melanocortin receptor 5, a newly identified ACTHR), and GR (glucocorticoid receptor). These nine genes will be used in the qRT-PCR validation and detection.

### 2.3. Analysis of Metabonomic Data

A total of 10,925 metabolites (5283 positive and 5462 negative) were obtained, and through the identification analysis and comparison, we identified 417 differential metabolites (DMs) (248 positive DMs and 169 negative DMs, 215 up-regulated and 202 down-regulated) (Appendix A) in three comparison groups. Among these groups (Figure 7), 70 DMs were increased and 117 were decreased in the ME_EBr_vs_ME_CBr group; 125 were increased and 79 were decreased in the ME_EHt_vs_ME_CHt group; and in the ME_EAG_vs_ME_CAG group, 42 were increased and 44 were decreased. All DMs were mapped to the HMDB to finish the statistical analysis (Appendix A). Most of the DMs were lipids and lipid-like molecules. All DMs were subjected to expression comparison analysis and KEGG pathway enrichment analysis, and these results are shown in Figure 8 and Figure 9.

### 2.4. Correlation Analysis Between the Transcriptomic and Metabonomic Data

The results of the correlation analysis showed that 61 DMs and 63 DEGs had significant potential associations in the brain group (Figure 10 Br_group); 64 DMs and 15 DEGs had significant potential associations in the adrenal gland group (Figure 10 AG_group); and 117 DMs and 2 DEGs had significant potential associations in the heart group (Figure 10 Ht_group). Through the further analysis, 19 DMs and 12 DEGs in the brain group had the regulatory association in important pathways (Figure 11A), and in the heart group, there were 32 DMs and 1 DEG that had the regulatory association in important pathways (Figure 11B). Among them, 17 DMs and 7 DEGs were up-regulated and 34 DMs and 6 DEGs were down-regulated (Figure 11C). The main affected pathways included the FoxO signaling pathway, the MAPK signaling pathway, arachidonic acid metabolism, and purine metabolism.

### 2.5. The qRT-PCR Results

As shown in Figure 12, CRHR2 was up-regulated in the brain (Figure 12A), while other genes showed no change (Figure 12A–C). This suggested that the transcriptome data of this RNA-seq was reliable. In the short-term noise stimulation group, the relative expression level of four genes (CRH, AVT, AVPR2, and GR) showed significant differences in brain tissue (Figure 12a), and five genes (CHRBP, AVPR2, MC5R, GR, and MC2R) showed significant differences in heart tissue (Figure 12b). Only two genes (MC2R and CRHBP) showed significant differences in adrenal gland tissue (AG) (Figure 12c). In dead fish, four genes (CRH, CRHR2, AVT, and POMC) showed significant differences in the brain (Figure 12d), three genes (CRHBP, AVPR2, and MC5R) showed significant differences in the heart (Figure 12e), and four genes (CRHR2, CRHBP, MC2R, and GR) showed significant differences in the AG (Figure 12f).

## 3. Discussion

In this study, we found that exposing *L. polyactis* to the engine noise of a small fishing boat for 6 days could cause death (Table 1) and induced symptoms of cerebral hemorrhage in the dead fish (Figure 3). These results were similar to those reported previously for noise stimulation in aquatic mammals [20,21,22]. It was likely that the underwater noise stimulation induced cardiovascular and cerebrovascular diseases in the small yellow croaker, but the underlying causes need to be further studied. The HPA axis plays a key role in the CNS during chronic stimulation. The dysregulation of this axis will cause many adverse consequences, such as cardiovascular disease, which can further induce cerebral hemorrhage. Therefore, we speculated that perhaps the underwater noise stimulation affected a variety of functional tissues, mainly including those of the HPA axis (hypothalamus and adrenal gland) and the heart. Finally, we evaluated the functional organization (brain, heart, and adrenal gland) of fish that survived long-term noise stress using transcriptomic and metabonomic studies.

### 3.1. Strong Noise Severely Disturbed the Expression and Regulation of Important Genes

We found that intense noise stress had significant effects on the transcription levels of many potentially neuropathogenic genes in the brain. These genes included GFAP (glial fibrillary acidic protein), PTPRT (receptor-type tyrosine-protein phosphatase T), ADAMTS (a disintegrin-Like and metalloproteinase with thrombospondin type 1 motif), BUB1 (budding uninhibited by benzimidazoles 1), and others that have been studied in a variety of neurological disorders. GFAP expression levels increased, which is a common feature of CNS injury. This increase is used as a marker to investigate mechanisms producing gliosis [37]. PTPRT and ADAMTS are important genes related to spinal cord injury and neurological diseases [38,39]. BUB1 was found to be highly expressed in the brain tissue of Alzheimer’ s disease (AD) patients and is a promising gene signature for diagnosis and therapy of AD [40]. In zebrafish, the down-regulation of EMILIN-1 (elastin microfibril interface-located protein 1) was shown to cause developmental delay, motor defects, and abnormalities in axonal branching of spinal motor neurons, and it could affect the pathogenesis of diseases in the peripheral nervous system [41]. Regulating the expression of FOSL2 (Fos-related antigen 2) may favor inflammation inhibition and hematoma resolution in intracerebral hemorrhage [42]. These results suggested that when *L. polyactis* was exposed to the noise stress environment for an extended time, the brain suffered the most serious consequences, even when the fish survived.

TG (thyroglobulin) is the protein precursor of thyroid hormones and is essential for growth, development, and the control of metabolism in vertebrates [43]. In our results, TG was down-regulated in brain and adrenal gland tissues, suggesting that long-term noise stress had affected the endocrine regulation of *L. polyactis*. In the heart group, there was only one unique DEG, which was PLA2G4. According to reports, PLA2G4 affects heart function by regulating the metabolic level of arachidonic acid (AA) and plays an important role in the development of myocardial injury and cardiovascular diseases [44,45,46,47]. Increased PLA2G4 expression has been found in many inflammatory diseases of different organs and tissues, including the brain, the heart, and atherosclerotic artery walls [45,46,48,49], suggesting that long-term noise stress could affect the heart and cardiovascular function of *L. polyactis*.

The results of the HPA axis evaluation had a surprising finding. Many genes related to HPA axis function were found in each group, but only CRHR2 was a DEG in the brain group (Table 2). This may imply that CRHR2 was an essential gene when HPA axis regulation was at a steady state. In the regulation pathways of the HPA axis, CRHR2 plays an important role in neuroendocrine and autonomic nerves of the hypothalamus, and in behavioral regulation. In a recent study [50], the researchers found that a deficiency of CRHR2 caused HPA axis dysfunction and induced anxiety in zebrafish. They found that in the Crhr2-null zebrafish, the deficiency of CRHR2 decreased cortisol levels, up-regulated crhr1, and down-regulated crhb, crhbp, ucn3l, and POMC. Additionally, the expression levels of TSPO (translocator protein, a neuro-inflammatory marker) and its ligand protein gene vdac1 were up-regulated. In an earlier study [51], they found that CRHR2 was involved in regulating the HPA axis stress response in Crhr2-null mice, and they speculated that it might affect blood pressure and the cardiovascular steady state. Thus, based on these research conclusions, it seems that the high expression of CRHR2 was required to maintain the regulatory function of the HPA axis during long-term underwater noise stress in the surviving *L. polyactis*.

However, the expression of many HPA-axis functional genes showed no significant change in the surviving *L. polyactis* exposed to long-tern noise stress, but did show significant change in the fish exposed to short-term noise stress (Figure 12a–c) and the dead fish (Figure 12d–f). According to previous research, external stress stimuli prompt the rapid response of the HPA axis in animals. In this feedback regulation process, when the external stimulus signal is received by the sensory organs in the body, the HPA axis is activated, and the hypothalamus begins to secrete CRH and AVT, which stimulate the pituitary to hydrolyze POMC to produce ACTH (adrenocorticotropic hormone) and alpha-MSH (alpha-melanocyte-stimulating hormone). Then, ACTH further promotes the adrenal gland to secrete GC to regulate the body’s development, growth, metabolism, immune function, etc. [29]. Meanwhile, increased concentrations of GC in the blood can inhibit the secretion of CRH and ACTH by the hypothalamus and pituitary, providing a negative feedback regulation mechanism [52,53,54,55]. There are also many other signature genes involved in HPA axis regulation, including CRHR2 [51], AVPR2 (activated by AVP/AVT and regulating physiological processes, including water homeostasis in the body) [56], MC2R (also called ACTHR, specifically combined with ACTH) [57], MC5R (which is the last melanocortin receptor identified and is widely expressed in both CNS and the peripheral organ system) [58], CRHBP (which mediates the reaction between CRH and CRHRs) [59], and GR (the GC receptor is a member of the nuclear receptor family and is involved in the regulation of various physiological and biochemical functions, including development, metabolism, inflammation, the stress response, and cardiovascular system steady-state regulation) [60]. Therefore, we speculated that HPA axis regulation may be in a state of negative feedback regulation in the fish exposed to short-term noise stress. In the brain (Figure 12a), CRH, AVT, and AVPR2 were down-regulated, but GR was up-regulated, possibly indicating the feedback regulation of GC (the high level of GR implied a high level of GC, and GC negatively regulated the expression levels of CRH, AVT, and ACTH). Meanwhile, the adrenal gland and heart are the downstream effector organs in the process of HPA axis regulation; therefore, during the feedback regulation of GC, signature genes in these tissues also showed similar expression changes (Figure 12b,c). However, in the dead fish exposed to long-term noise stress, the negative feedback regulation of GC was not activated in the brain, so CRH, CRHR2, AVT, and POMC remained at high expression levels (Figure 12d) and the signature genes of the downstream effector organs had high expression levels (Figure 12e,f). Therefore, it was implied that the cause of death in *L. polyactis* exposed to noise stress was likely related to the regulation of the HPA axis. In conclusion, for the surviving *L. polyactis* in the long-term noise stimulation group, the transcriptome data analysis showed that they were at sub-optimal health levels, and although not immediately lethal, the noise stress caused chronic health problems that could lead to death.

### 3.2. Strong-Noise-Induced Metabolic Disorders

Through the analysis of transcriptome data and the expression analysis of genes related to the HPA axis, we seemed to prove that strong noise stress (>120 dB re 1 µPa) can induce death in the small yellow croaker by affecting the regulation of the HPA axis. However, the metabolic changes in vivo still needed to be further analyzed to see whether the body could adapt to noise stimulation and continue to survive. In the metabonomic data, the three comparison groups had many DMs (Figure 7 and Figure 8), suggesting that long-term noise stress can significantly affect a variety of biological pathways and aspects of molecular metabolism.

First, we found some representative hormones that have a key role in the regulation of the HPA axis, such as cortisol (a GC) (up-regulated, Figure 8A), neurotensin (a neuropolypeptide hormone with significant antihypertensive effects) (down-regulated, Figure 8B), and isoproterenol (down-regulated, Figure 8C). Through the KEGG pathway enrichment analysis of DMs (Figure 9), we found that GPL metabolism was a common and significant enrichment pathway in all three groups. GPLs are the primary building blocks of cellular membranes and play an important role in a variety of cellular signal transduction pathways; thus, their composition and abundance have an important effect on the stability of the internal environment, and can reflect a variety of pathological states [61]. GPLs include a variety of phospholipids of different composition, such as PC, phosphatidylethanolamine (PE), phosphatidylserine (PS), phosphatidylinositol (PI), and phosphatidic acid (PA). In total, 46 significantly differentially expressed phospholipids were identified in our study (Appendix A), including 29 PCs, 12 PEs, 4 PSs, and 1 PA. Many studies indicated that many lipids are enriched in the brain, and most of them are GPLs. Once the brain is injured, GPL metabolism is significantly altered, thus becoming an indicative characteristic of many brain diseases, such as traumatic brain injury (TBI) [61], Alzheimer’s disease (AD) [62,63], and post-stroke depression (PSD) [64]. Due to the diversity of GPLs, they are also important functional metabolites in other tissues of body. Some studies have shown that GPL metabolism is involved in the repair of myocardial infarction and plays a crucial role in the occurrence and development of cardiovascular diseases [65]. Another study reported that various stress responses in fish can lead to metabolic disorders, such as the phosphate-stress-induced disruption of lipid metabolism, GPL metabolism, purine metabolism, and the tricarboxylic acid (TCA) cycle [66] in *Scophthalmus maximus*. In addition to GPL metabolism, there were many other important metabolic pathways that showed significant differences in our study, including pantothenate and CoA biosynthesis, the GnRH signaling pathway, purine metabolism, nucleotide metabolism, and the TCA cycle. The findings indicated that long-term noise stress interfered with important metabolic pathways, and such disturbances were likely to induce a variety of neurological or endocrine-related diseases.

### 3.3. The Processes and Causes of Death Induced by Loud Noise

In order to better analyze the processes and causes of death caused by strong underwater noise stimulation, we explored the correlation between the changes in transcription levels and metabolic levels in *L. polyactis* under such stimulation. In our results (Figure 10 and Figure 11), we found that long-term noise stress had a significant impact on the brain and heart. A total of 19 DMs and 12 DEGs in the brain group had significant regulatory associations with oxidative phosphorylation, the FoxO signaling pathway, the MAPK signaling pathway, etc. (Figure 11A). In the heart group, 32 DMs and 1 DEG had significant regulatory associations with the MAPK signaling pathway, arachidonic acid metabolism, GLP metabolism, and others (Figure 11B). Most of these pathways were related to the body’s antioxidant system, DNA repair, lipid metabolism homeostasis, etc., and the activation starting points of these pathways included INS (insulin), IGH (immunoglobulin heavy chain), ATP5F1 (ATP synthase F(0) complex subunit B1), COX6A (cytochrome c oxidase subunit 6A), cortisol, and AVT. There were also close connections between them. Cortisol is a GC, and like AVT, is one of the representative hormones in the HPA axis; meanwhile, INS can be affected by GCs in the regulation pathways of the HPA axis [67].

In the cGMP-PKG signaling pathway, INS can activate GCs through the PI3K-Akt signaling pathway, affecting the synthesis of cGMP, which will activate PRKG1 (cGMP-dependent protein kinase I) to affect downstream reactions, including the MAPK signaling pathway, and purine metabolism (Figure 11A). Moreover, some researchers have found that the mitochondria can sense GC in order to mediate the stress response; however, the excessive and dysregulated ATP metabolism in mitochondria can cause homeostasis imbalances in purine catabolism and oxidative stress, causing damage to tissues and organs [68,69,70]. Our results were similar to these conclusions. ATP5F1 and COX6A were up-regulated in the Br_group (Appendix A); these are mitochondrial function genes, involved in the synthesis of SOD (superoxide dismutase) and activating the antioxidant system. Additionally, DMs, ADP, ADP-ribose, and adenosine 3′ 5′-diphosphate (Appendix A) are intermediates of the oxidative phosphorylation process and energy metabolism in purine metabolism (Figure 11A). Therefore, we speculated that long-term noise stress may have caused transcriptional and metabolic disorders in the brain, and further induced oxidative stress damage.

In the heart, INS and AVT are associated with changes in blood pressure, and they are the key regulators of cardiovascular disease [53,71]. In our results, PLA2G4 was the only DEG in the heart group. This implied that there was a significant association between INS, AVT, and PLA2G4. Through further analysis, we found that INS could affect the expression of PLA2G4 through the insulin signaling pathway and MAPK signaling pathways (Figure 11B). AVT also effected the expression of PLA2G4 in vascular smooth muscle contraction (Figure 11B). Moreover, PLA2G4 can hydrolyze PCs to produce LysoPCs, which can generate ROS (reactive oxygen species) and eventually damage cells and impair mitochondrial function. Therefore, the long-term noise stress had a serious effect on lipid metabolism in the heart, and this effect was likely achieved by interfering with the regulation of the HPA axis to affect PLA2G4.

Therefore, based on the above results and analyses, we suggested that long-term noise stress could significantly disturb the interactions of multiple important signaling pathways and metabolic functions by affecting the regulation of the HPA axis, resulting in oxidative stress damage to tissues and leading to eventual death. However, when exposed to short-term noise stress, the activity of the HPA axis of *L. polyactis* decreased significantly and tended to be stable, so there was no significant effect. In brief, strong noise can seriously affect the survival of *L. polyactis*, especially when persisting for a long time, and this effect is often fatal.

## 4. Materials and Methods

### 4.1. Animals

The experimental small yellow croakers were from an artificially bred F8 generation in Xiangshan County, Ningbo, China. In 2014, the first artificial breeding experiment was completed in the Xiangshan harbor aquatic seeding Co., Ltd. (Xiangshan County, Ningbo, China). In this study, we selected 150 fishes for the underwater noise stress experiment. These fishes consisted of 75 males and 75 females, with a body weight of approximately 40–60 g and a body length range of 12–14 cm. Before the experiment, all fish were raised in a 2000 L breeding bucket in the culture house (water temperature: 14–16 °C, in May 2023). And to avoid the death by unknown factors, the feeding was stopped one day before the experiment, and no feeding took place during the experiment. These preparations were completed in May 2023.

### 4.2. Noise Stress Experiment

In our study, we constructed a set of underwater noise stress models (Figure 1A), including an underwater sound broadcast system, an underwater sound acquisition system, and a video monitoring system. The underwater sound broadcast system had two parts, a UW30 amplifier (EV, South Bend, IN, USA) and a UX70 audio power amplifier (SANSUI, Shenzhen, China). The underwater sound acquisition system (Appendix A) included an HTD42 hydrophone (sensitivity @1kHz: −207.5dB V/μPa ± 2dB) (TD, Beijing, China), an audio data collector (TD, Beijing, China), and three operational software systems (Server_Monitor v1.0.4 software, WavConverter v1.0 software, and Audacity v3.4.2) (TD, Beijing, China). The video monitoring system included a monitor (HIKVISION, Hangzhou, China) and a storage hard disk (HIKVISION, Hangzhou, China) used for the real-time monitoring and video preservation of the underwater environment.

A diagram showing the setup of the noise stress experiment is shown in Figure 1B. The experimental group was cultured in a 1200 L breeding bucket in a single room, and the control group was cultured in three 1000 L breeding buckets in another room. Every breeding bucket contained 30 fishes. Fishes were acclimated to the new barrel for 1 h before the experiment began. We selected the engine noise of a small fishing boat (a common ship in aquaculture areas) as the noise stimulus source. In the control group, no additional stimulus sound source was added to the background sounds coming from the oxygen supply equipment. The long-term underwater noise stress experiment lasted for 6 days, and the short-term underwater noise stress experiment lasted for 10 min. The two experiments were performed separately.

After completing each experiment, we collected the brains, hearts, and adrenal glands of the dead and surviving fish (we carefully observed the status of the fish using the video monitoring system and collected the dead fish immediately). All tissue samples were collected into 2 mL EP tubes using tweezers, rapidly frozen with liquid nitrogen, and transferred to a −80 °C freezer for storage. In the end, a total of 58 fishes were collected, including 34 from the long-term noise stress group (stimulation group: 17; control group: 17), 12 from the short-term noise stress experiment (stimulation group: 6; control group: 6), and 8 dead fishes (stimulation group: 8). These experiments and the sample collection works were completed at the Xiangshan harbor aquatic seeding Co., Ltd. (Xiangshan County, Ningbo, China) in May 2023.

### 4.3. Total RNA Isolation and Illumina Sequencing

Some samples (the brain, heart and adrenal gland) of surviving fishes in the long-term noise stress group were used for RNA-seq (stimulation group: 3 fishes; control group: 3 fishes). The extraction of total RNA was performed using QIAzol Lysis Reagent (Qiagen, Dusseldorf, Germany), and then the NanoDrop 2000 Spectrophotometer (Thermo Fisher Scientific, Cheshire, UK) was used to detect the RNA quantity, and agarose gel electrophoresis to detect the RNA integrity. Subsequently, the Agilent 5300 (Agilent, Santa Clara, CA, USA) was used to measure the RIN number. For RNA-seq (RNA Sequencing), first, oligo (dT) magnetic beads were used to perform the poly (A) enrichment of total RNA, and then approximately 1 μg (total RNA) was used to prepare an sRNA library according to the protocol of the Illumina^®^ Stranded mRNA Prep Kit (Illumina, San Diego, CA, USA). The RNA-seq was conducted using the Illumina Novaseq 6000 platform (Majorbio, Shanghai, China).

### 4.4. Alignment of Transcriptomic Data

First, we analyzed the base mass, base error rate, and base content of raw sequences by filtering out the low-quality sequences. Second, we selected the high-quality sequences to map onto the *Larimichthys polyactis* reference genome [36] using TopHat2 v2.1.1 and HISAT2 v2.0.5 [72,73]. Third, we used Cufflinks v0.8.0 software to complete the mapped reads’ assembly and function annotation [74]. The total reads number of every gene was generated using RSEM v1.3.3 software [75]; meanwhile, DESeq2 v1.42.0 software was used to analyze the expression level of differential genes [76], and lastly, the FDR (false discovery rate) value ≤ 0.05 and the absolute value of log2 (fold change) ≥ 1 were used as the criterion to assess the statistically significant DEGs.

### 4.5. Functional Annotation and Pathway Enrichment Analysis

After alignment and analysis, the obtained genes will translated to protein sequences, and then DIAMOND v2.1.8 (https://github.com/bbuchfink/diamond (accessed on 12 July 2023)) was used to complete BLAST in SwissProt [77]; meanwhile, all these genes and their protein sequences were also subjected to BLAST in GO, COG (Clusters of Orthologous Groups of proteins), and KEGG, to complete the functional annotation analysis. On the other hand, for the pathway enrichment analysis of every mapped genes, we used Goatools v0.7.6 (https://github.com/tanghaibao/GOatools (accessed on 12 July 2023)) to finish the GO term enrichment analysis [78], and the R package clusterProfiler v4.0 to finish the KEGG pathway enrichment analysis [79].

### 4.6. Metabolite Extraction and UPLC-MS/MS Analysis

We selected the brain, heart, and adrenal gland of 12 surviving fishes (stimulation group: 6; control group: 6) from the long-term noise stress group for the metabolite extraction. Every tissue sample was cut into 50 mg of solids to transfer to 2 mL centrifuge tubes, then a 6 mm diameter grinding bead and 400 µL of extraction solution (methanol/water = 4:1 (*v*/*v*) were added, containing 0.02 mg/mL of internal standard (L-2-chlorophenylalanine)) for metabolite extraction. Next, the Wonbio-96c frozen tissue grinder (Shanghai wanbo biotechnology Co., Ltd. #32, Xinxin Road, Chedun Town, Songjiang District, Shanghai, China) was used to grind the samples for 6 min (−10 °C, 50 Hz), followed by low-temperature ultrasonic extraction for 30 min (5 °C, 40 kHz). After grinding and extraction, the samples were left at −20 °C for 30 min and then centrifuged for 15 min (4 °C, 13,000× g), and, at last, the supernatant was transferred into the injection vial and a SCIEX UPLCTriple TOF 5600 system which was equipped with an ACQUITY HSS T3 column (100 mm × 2.1 mm i.d., 1.8 µm; Waters Corporation, Milford, MA, USA) was used to start the LC-MS/MS analysis. Therein, the mobile phases consisted of 0.1% formic acid in water/acetonitrile (95:5, *v*/*v*) (solvent A) and 0.1% formic acid in acetonitrile/isopropanol/water (47.5:47.5, *v*/*v*) (solvent B), the flow rate was 0.40 mL/min, and the column temperature was 40 ℃. This work was completed in Majorbio Bio-Pharm Technology Co., Ltd. (Shanghai, China).

### 4.7. Data Analysis and Statistics of Metabolites

The Progenesis QI v2.3 (Waters Corporation, Milford, MA, USA) software was used to analyze and evaluate the metabolite raw data, and a three-dimensional data matrix was generated in CSV format. After filtering and screening, all the metabolites were identified in the main databases, including Metlin (https://metlin.scripps.edu/ (accessed on 4 September 2023)), HMDB (http://www.hmdb.ca/ (accessed on 4 September 2023)), the Majorbio Database, and so on. And all these data were analyzed by the free online platform of the Majorbio cloud platform (cloud.majorbio.com (accessed on 4 September 2023)) [80]. Then, following online analysis to retain the samples which had at least 80% metabolic features, at last, minimum metabolite values were imputed for specific samples in which the metabolite levels fell below the lower limit of quantitation, and each metabolic feature was normalized by sum. Furthermore, we removed the variables with a relative standard deviation (RSD) > 30% of QC samples, and performed log10 processing to obtain the final data matrix for subsequent analysis, after data preprocessing, to perform the variance analysis on the matrix file.

### 4.8. Differential Identification and Pathway Enrichment Analysis of Metabolites

After data analysis and statistics, we used the R package “ropls” v1.6.2 to perform the principal component analysis (PCA), orthogonal least partial squares discriminant analysis (OPLS-DA), and 7-cycle interactive validation. Then, we used the OPLS-DA model to obtain the VIP (variable importance in the projection), and performed Student’s *t*-test to obtain the *p*-value. At last, the metabolites with VIP > 1, *p* < 0.05, were determined to be DMs. Then, based on the KEGG database (http://www.genome.jp/kegg/) (accessed on 6 September 2023), these DMs were mapped onto their biochemical pathways using metabolic enrichment and pathway analysis. In addition, we accorded to the pathway and function of the metabolites to classify them, and used the Python package “scipy.stats” v1.11.4 (https://docs.scipy.org/doc/scipy/) (accessed on 6 September 2023) to perform enrichment analysis to obtain the most relevant and focused biological pathways.

### 4.9. QRT-PCR Validation and Detection

We selected 8 surviving fishes of the long-term noise stimulation group, 8 fishes of the short-term noise stimulation group, and 4 dead fishes of the long-term noise stimulation group to collect the brain, heart, and adrenal gland for the qRT-PCR validation and detection. All the primers used for qRT-PCR are listed in Appendix A. Using the Triol method (Vazyme, Nanjing, China), we isolated total RNA, and after a quality test, we used the Hifair^®^ II 1st Strand cDNA Synthesis Kit (Yeasen, Shanghai, China) to synthesize the first strand of cDNA; all this cDNA was stored at −80 °C until use. Next, we used the SYBR^®^ Premix Ex Taq™ II Kit (Takara, Kyoto, Japan) to finish the preparation of the reaction solution, and then, using the ABI 7500 qPCR instrument (Thermofisher, Sunnyvale, CA, USA), we carried out testing. β-actin was used as the reference gene, and we calculated the relative mRNA expression levels by using the comparative Ct (2^−∆∆Ct^) method [81].

## 5. Conclusions

This study was the first to use an underwater noise stress model to simulate the effects of controlled intense underwater noise pollution on the HPA axis and lethal threat of *L. polyactis*. We found that long-term stimulation with the engine noise of a small fishing boat induced the death of *Larimichthys polyactis*, and the dead fishes had brain hemorrhage symptoms. Further research showed that the long-term noise stress affected the regulation of the HPA axis to cause the metabolic disorders in the body, including purine metabolism, GPL metabolism, and ether lipid metabolism, then induced changes in a variety of biological signaling pathways, and may have further induced body oxidative stress injury and the occurrence of other related diseases, such as cardiovascular and cerebrovascular diseases. And the causes of such long-term noise stress death may be divided into two scenarios: (1) the regulation of the HPA axis appeared overactive or unbalanced, and caused the excessive stress response; finally, it directly caused death; (2) the regulation of the HPA axis was in a state of dynamic balance, but the internal environment metabolism had been disordered, and if this disordered state persisted, it subsequently caused chronic death.

## Figures and Tables

**Figure 1 ijms-25-12610-f001:**
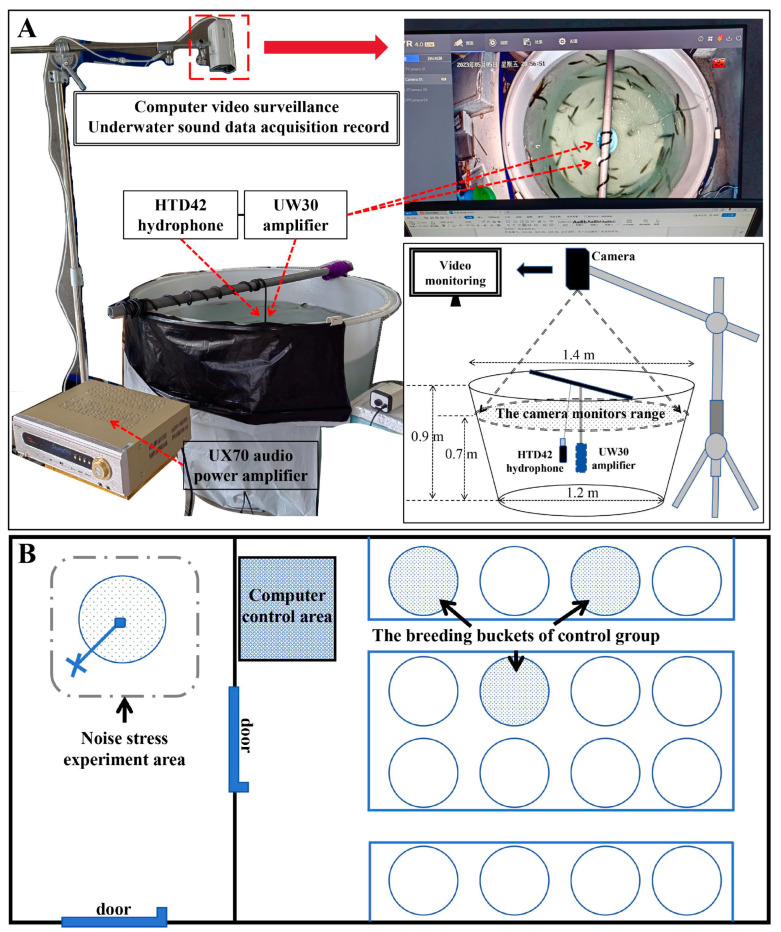
The underwater noise stress model and the plane distribution of the experimental area. (**A**) The underwater noise stress model; (**B**) the plane distribution of the experimental area, including noise stress experiment area, computer control area, and control group area.

**Figure 2 ijms-25-12610-f002:**
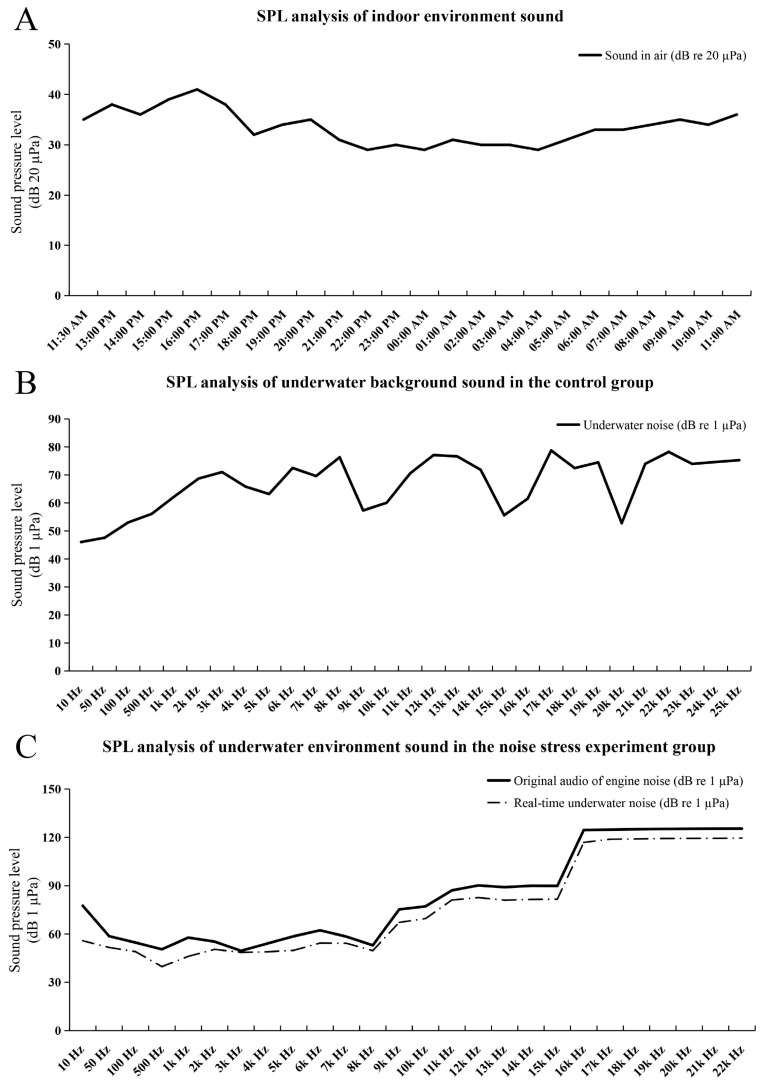
SPL of background noise. (**A**) SPL analysis of indoor environment sound. The abscissa represents the detection time, and the ordinate represents SPL (dB 20 µPa); (**B**) SPL analysis of underwater background sound in the control group; when the oxygen valve is opening, this is the background noise. The abscissa represents the frequency (10–25,000 Hz), and the ordinate represents SPL (dB 1 µPa); (**C**) SPL analysis of underwater environment sound in the noise stress experiment group, including the original audio of engine noise and the real-time underwater noise. The abscissa represents the frequency (10–22,000 Hz), and the ordinate represents SPL (dB 1 µPa).

**Figure 3 ijms-25-12610-f003:**
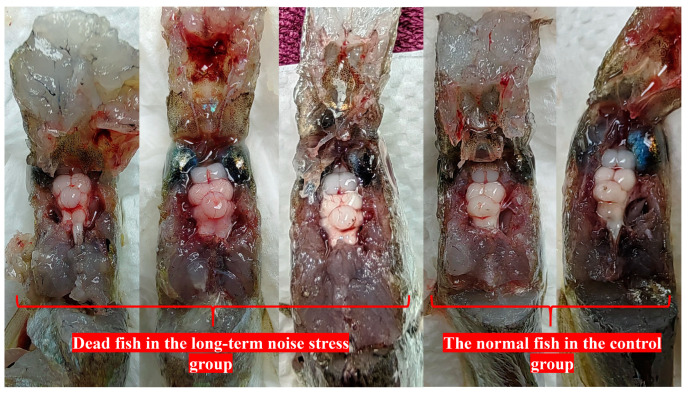
The brain changes in dead fishes in the long-term noise experiment. On the (**left**) side are three dead fishes from the stimulation group, and on the (**right**) side are normal fishes in the control group.

**Figure 4 ijms-25-12610-f004:**
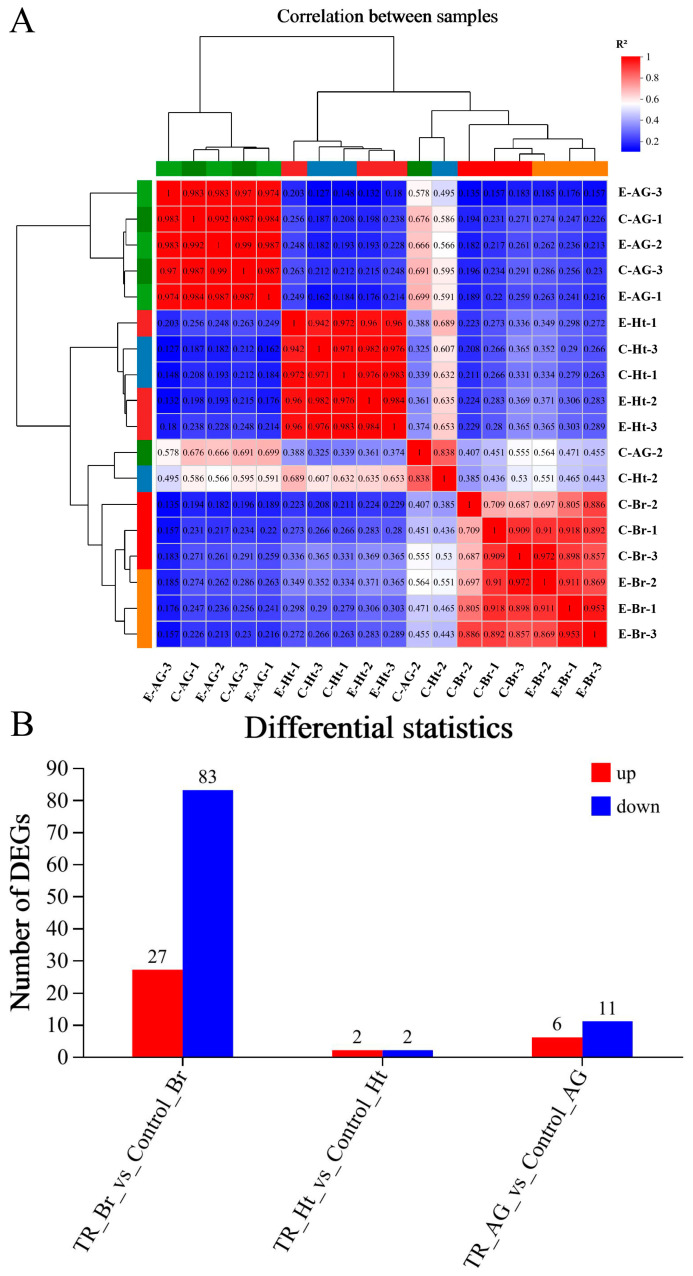
Transcriptomic data analysis. (**A**) The correlation analysis between biological replicate samples; (**B**) the differential gene comparisons between three groups. Note: “E” represents the experimental group and “C” represents the control group.

**Figure 5 ijms-25-12610-f005:**
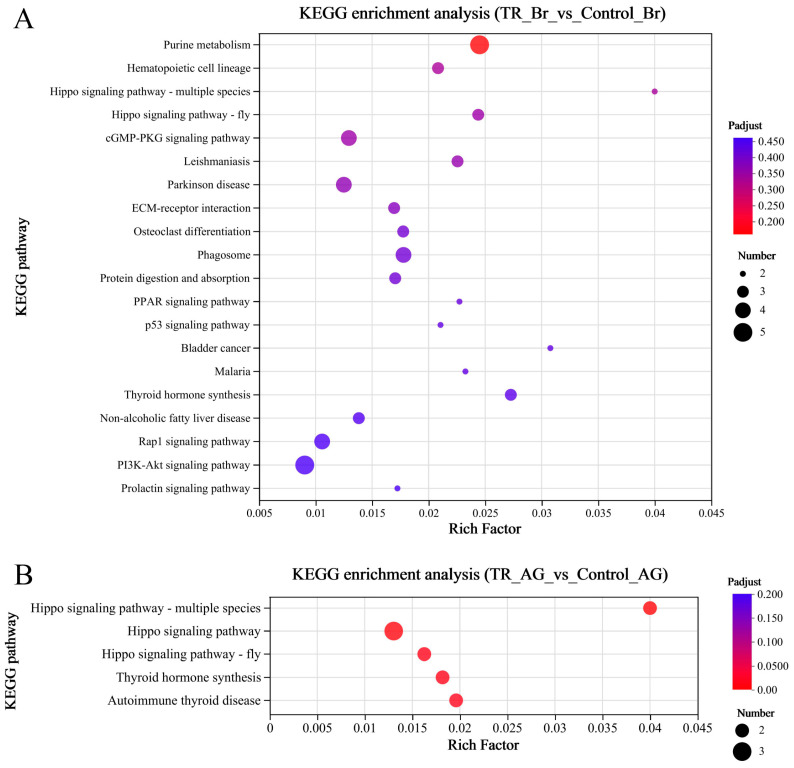
Scatterplot of the KEGG pathway enriched by differentially expressed genes. (**A**) The result of the KEGG pathway enrichment analysis in the DEGs of the brain group; (**B**) the result of the KEGG pathway enrichment analysis in the DEGs of the adrenal gland group.

**Figure 6 ijms-25-12610-f006:**
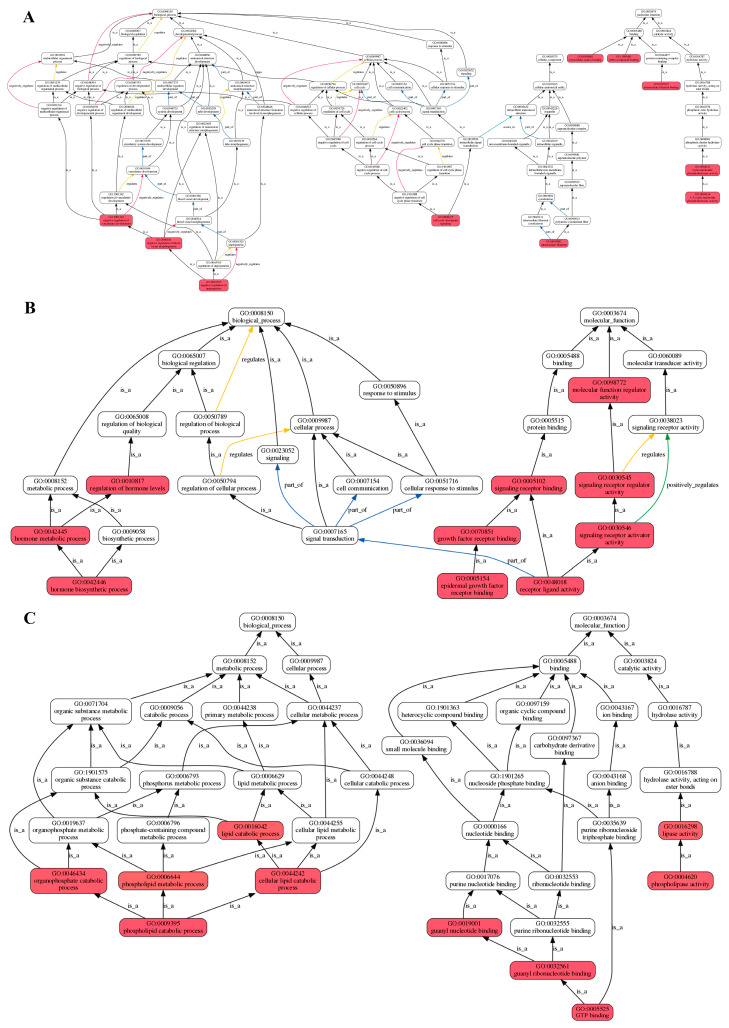
GO functional enrichment oriented analysis graph. (**A**) The result of the GO functional enrichment oriented analysis in the DEGs of the brain group; (**B**) the result of the GO functional enrichment oriented analysis in the DEGs of the adrenal gland group; (**C**) the result of the GO functional enrichment oriented analysis in the DEGs of the heart group.

**Figure 7 ijms-25-12610-f007:**
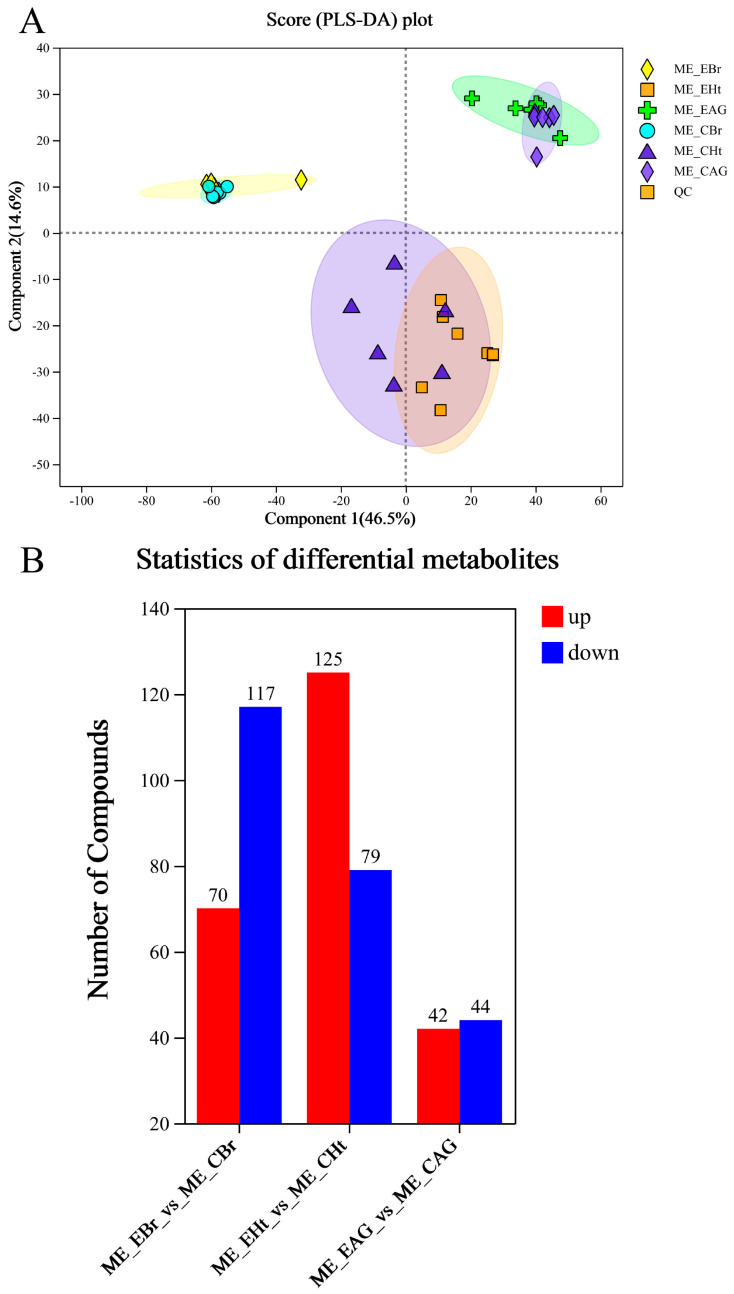
Transcriptomic data analysis. (**A**) The correlation analysis between biological replicate samples; (**B**) the differential gene comparisons between four groups.

**Figure 8 ijms-25-12610-f008:**
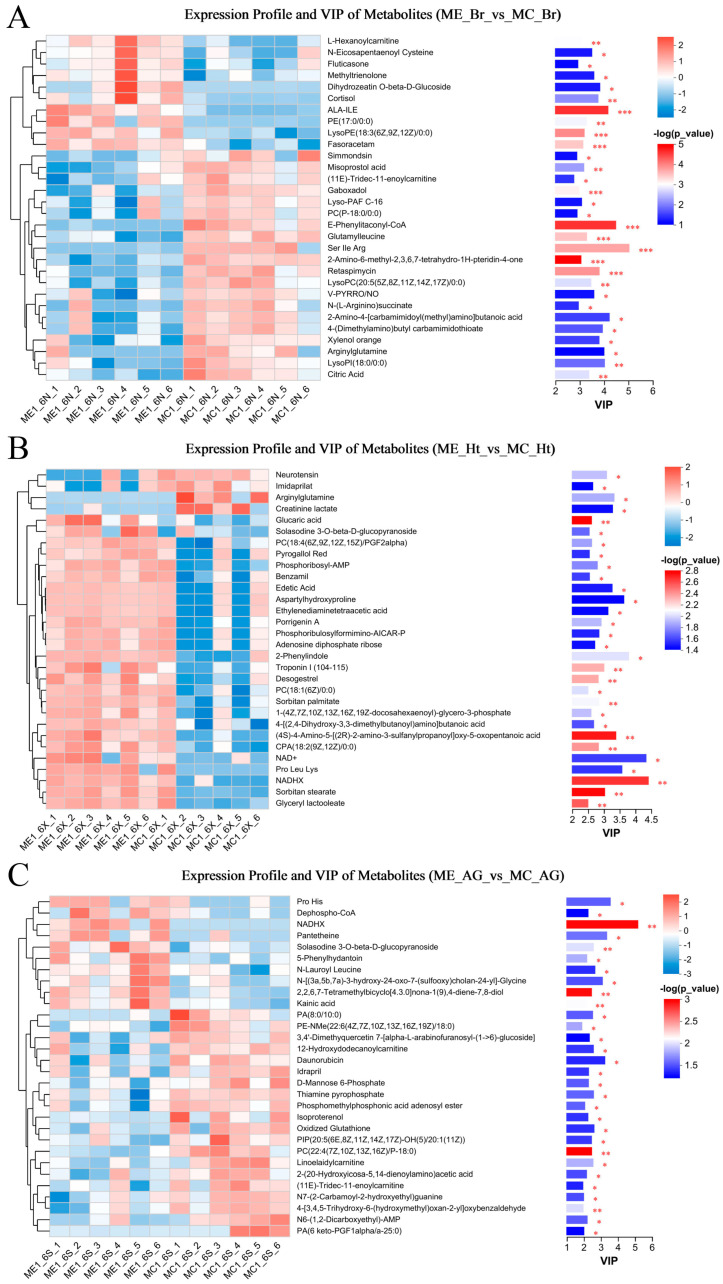
The top 30 differential metabolites’ expression profile and VIP analysis. (**A**) The result of the brain group; (**B**) the result of the heart group; (**C**) the result of the adrenal gland group. On the right is the VIP bar chart of the metabolite; the length of the bar indicates the contribution value of the metabolite to the difference between the two groups, and the larger value indicates the greater difference in the metabolite between the two groups. The color of the bar indicates the significance of the difference between the two groups of samples, that is *p*_value; the smaller the *p*_value and the larger the −log10 (*p*-value), the darker the color; the significance of differences is indicated by “*”. “*” represents *p* < 0.05, “**” represents *p* <0.01, and “***” represents *p* < 0.001.

**Figure 9 ijms-25-12610-f009:**
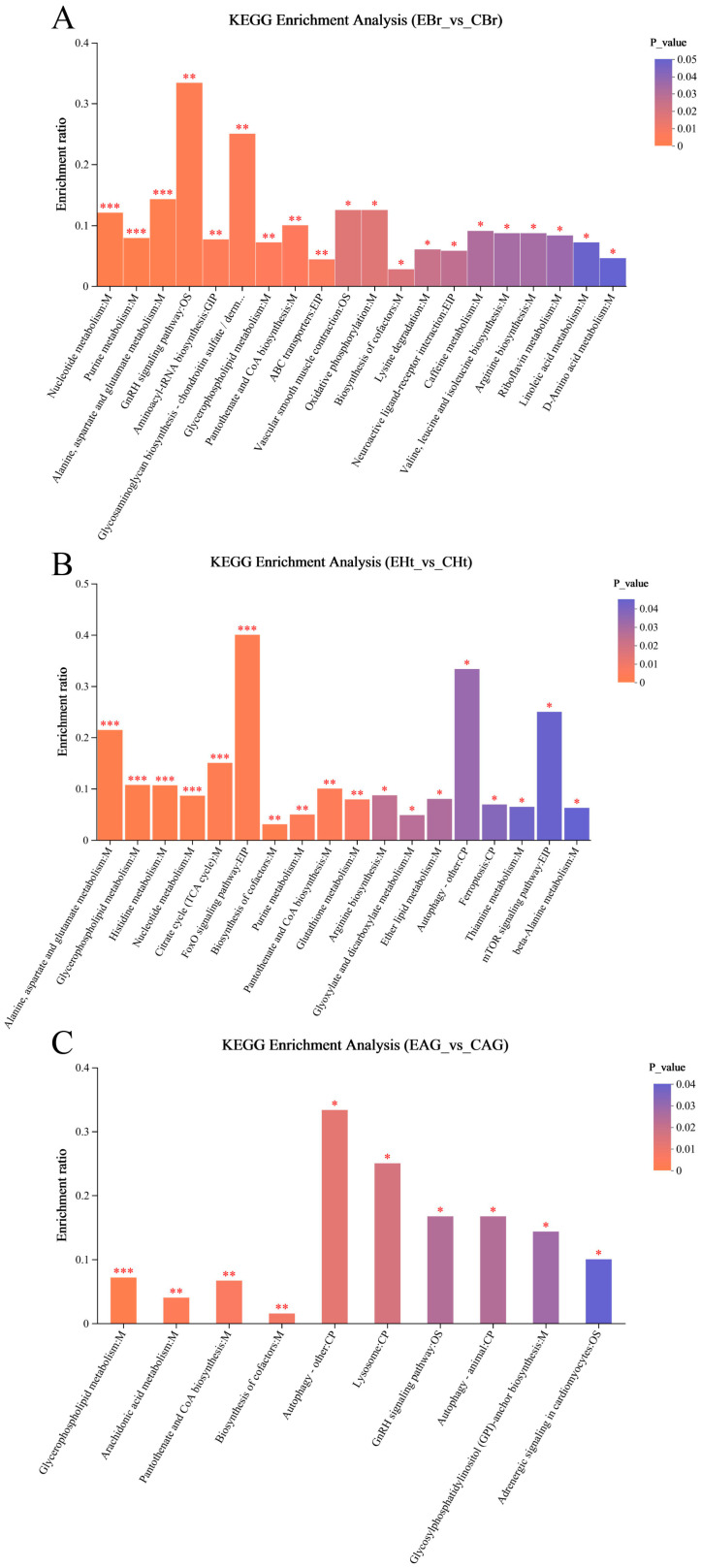
Bar graph of KEGG enrichment analysis of differential metabolites. (**A**) The result of the KEGG enrichment analysis in the DMs of the brain group; (**B**) the result of the KEGG enrichment analysis in the DMs of the heart group; (**C**) the result of the KEGG enrichment analysis in the DMs of the adrenal gland group. The ordinate is the enrichment ratio, and the abscissa is the metabolic pathway name; the color of the bar indicates the enrichment significance, using *p*_value to represent it, and the smaller the value, the more significant the enrichment. “*” represents *p* < 0.05, “**” represents *p* < 0.01, and “***” represents *p* < 0.001.

**Figure 10 ijms-25-12610-f010:**
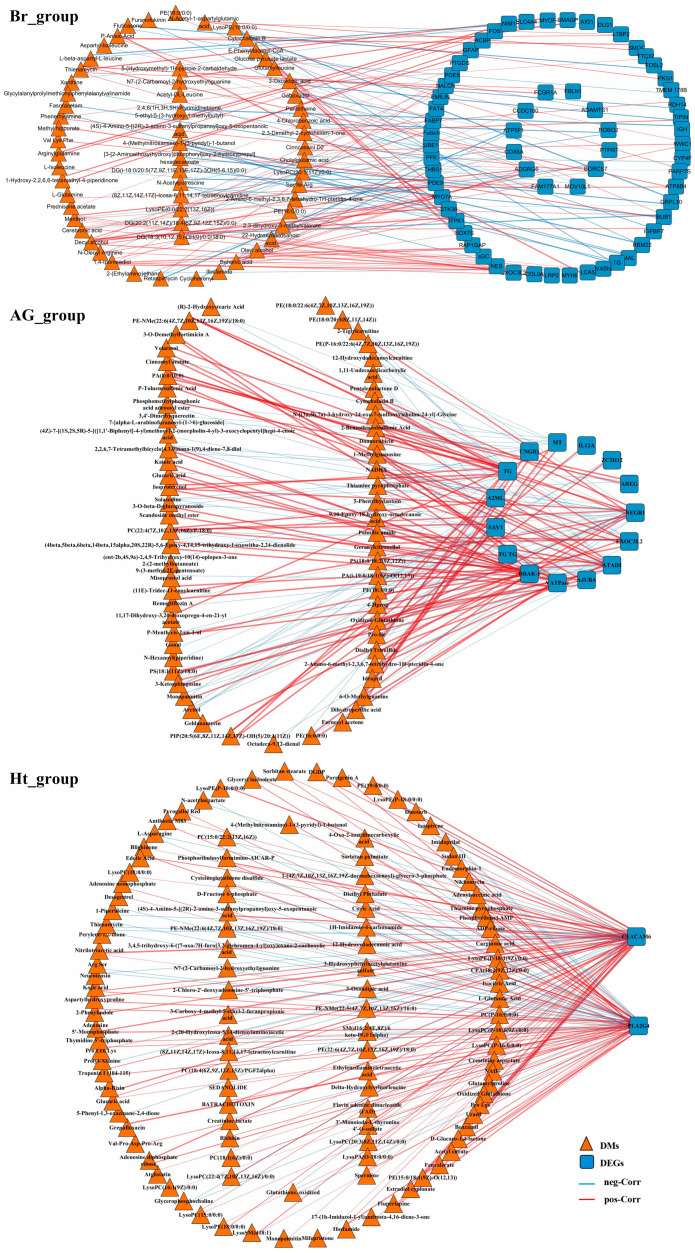
Positive (pos)/negative (neg) correlation analysis between DMs and DEGs. “Br_group”: the result of the correlation analysis between DMs and DEGs in the brain group; “AG_group”: the result of the correlation analysis between DMs and DEGs in the adrenal gland group; “Ht_group”: the result of the correlation analysis between DMs and DEGs in the heart group. An orange triangle represents a DM and a blue square represents a DEG; the blue line represents the negative correlation (neg-Corr) and the red line represents the positive correlation (pos-Corr).

**Figure 11 ijms-25-12610-f011:**
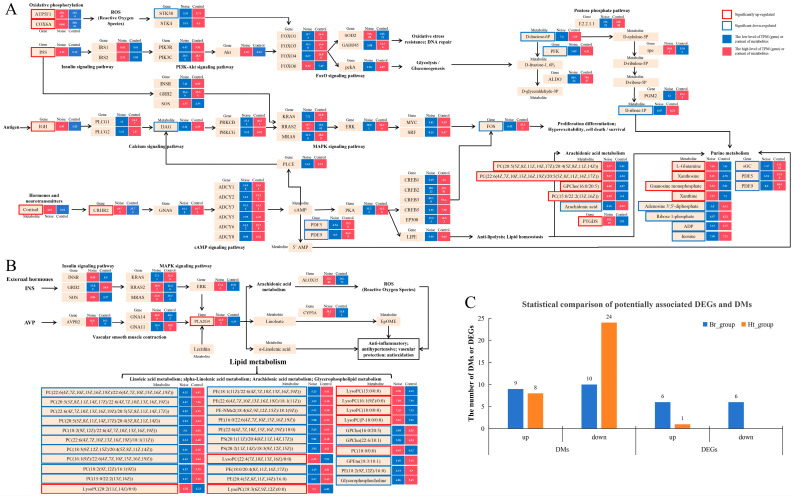
Correlation analysis of DMs and DEGs in signaling pathway and metabolic pathway. (**A**) The result of the correlation analysis in the DMs and DEGs of the brain group; (**B**) the result of the correlation analysis in the DMs and DEGs of the heart group; (**C**) the statistical comparison of potentially associated DEGs and DMs. A red box represents significantly up-regulated (*p* < 0.05), and a blue box represents significantly down-regulated metabolites/genes (*p* < 0.05); a blue square indicates a low level of TPM (gene) or content of metabolite in this sequencing result, and a red square indicates a high level of TPM (gene) or content of metabolite in this sequencing result.

**Figure 12 ijms-25-12610-f012:**
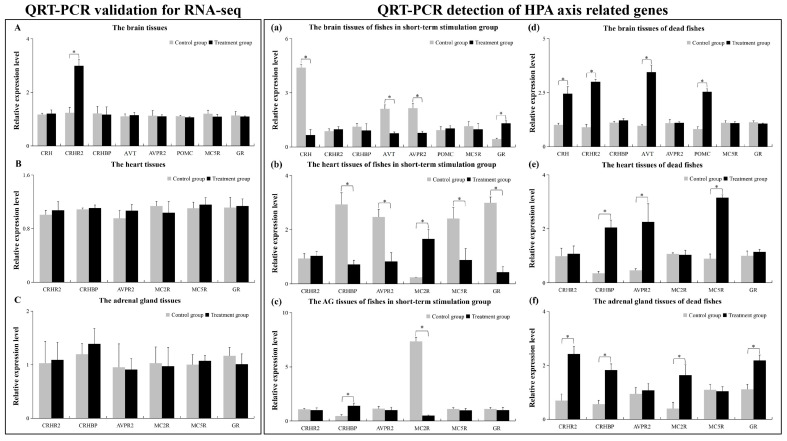
The qRT-PCR results. “QRT-PCR validation for RNA-seq”: qRT-PCR validation of the relative expression levels of HPA-axis-related genes in surviving fishes of the long-term noise stimulation group; (**A**) the brain tissues, (**B**) the heart tissues, (**C**) the adrenal gland tissues. “QRT-PCR detection of HPA-axis-related genes”: qRT-PCR detection of HPA-axis-related genes in the fishes of the short-term noise stimulation group and dead fishes of the long-term noise stimulation group; (**a**) the brain tissues of fishes in the short-term stimulation group, (**b**) the heart tissues of fishes in the short-term stimulation group, (**c**) the adrenal gland tissues of fishes in the short-term stimulation group; (**d**) the brain tissues of dead fishes, (**e**) the heart tissues of dead fishes, (**f**) the adrenal gland tissues of dead fishes. “*” (*p* < 0.05) represents the significance of the difference between the two groups.

**Table 1 ijms-25-12610-t001:** The information statistical records of dead fishes in the long-term noise stimulation experiment.

Experiment Type	Stimulation Time	Dead Fish/Total	Occurrence Time
Short-term stimulation group	10 min (start at 16:02 p.m. on 4 May 2023)	No deaths	/
Control group (short-term experiment)	no treatment	No deaths	/
Long-term stimulation group	6 days (start at 13:54 p.m. on 6 May 2023)	8/30	7 May 2023, at 9:17 a.m., 2 dead fishes; 7 May 2023, at 14:31 p.m., 1 dead fish; 7 May 2023, at 19:12 p.m., 1 dead fish; 7 May 2023, at 20:25 p.m., 1 dead fish; 8 May 2023, at 12:14 p.m., 1 dead fish; 8 May 2023, at 17:40 p.m., 1 dead fish; 10 May 2023, at 9:30 a.m., 1 dead fish
Control group (long-term experiment)	no treatment	1/30	May 10, 2023, at 7:28 a.m., 1 dead fish

**Table 2 ijms-25-12610-t002:** The expression difference statistics for 9 HPA-axis-related genes in three tissues.

Gene	Br Group	AG Group	Ht Group	Sequence_ID
*CRH*	no | down	/	/	evm.TU.Scaffold11.445
*CRHR2*	yes | up	no | down	no | down	evm.TU.Scaffold165.70
*CRHBP*	no | up	no | up	no | up	evm.TU.Scaffold2659.208
*AVT*	no | up	/	/	evm.TU.Scaffold2659.150
*AVPR2*	no | up	no | up	no | up	evm.TU.Scaffold218.13
*POMC*	no | up	no | up	no | up	evm.TU.Scaffold169.472
	no | up	no | down	no | down	evm.TU.scaffold20s2.1
*MC2R*	/	no | up	/	evm.TU.Scaffold95.252
*MC5R*	no | up	no | up	no | up	evm.TU.Scaffold95.251
*GR*	no | down	no | down	no | down	evm.TU.Scaffold306.164

Notes: “yes” and “no” represent significance of differential expression level; “up” and “down” represent the types of differential expression.

## Data Availability

The raw data of transcriptome have been uploaded to NCBI (project number: PRJNA1080343); the NCBI accession numbers are SRR28089698; SRR28089699; SRR28089700; SRR28089701; SRR28089702; and SRR28089703. And other data that support the findings of this study are available from the corresponding author upon reasonable request.

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
