# Peer review of "Metabonomics and Transcriptomics Analyses Reveal the Underlying HPA-Axis-Related Mechanisms of Lethality in Larimichthys polyactis Exposed to Underwater Noise Pollution"

_ijms, 2024, doi:10.3390/ijms252312610_

Round 1
Reviewer 1 Report
Comments and Suggestions for Authors
Please see the attached files.

Please see the attached files.
Please ask a native English speaker to proofread the manuscript. Many grammatical errors need to be corrected.
Author Response
Comments 1: Please ask a native English speaker to proofread the manuscript. Many grammatical errors need to be corrected.
33-40, 48-54, 60-64, 69-75, 86-90, 90-96,279-285, 330-333, 399-410, 413-417, 418-425, 450-458, 459-468, 496-500, 523-527, 542-547, 556-562, 585-592, 597-601, 603-609, 625-630, 650-654, etc, has a long sentence and grammatical errors. Please revise it for better understanding.
Response 1: Thank you very much for your advice, we have simplified and revised most of the long sentences, and conducted the English editing service of MDPI as a way to improve the quality of the manuscript. For details, please see the red marked section of the manuscript.
Comments 2: In NCBI, I cannot find your transcriptomic data PRJNA1080343. Make sure that you have successively uploaded the RNA-seq data to the NCBI and get a series of SRA data. This should have 18 SRA files. For example, we should get this information by search PRJNA1080343 in NCBI. Simlalr files:SRR8953110, SRR8953105, SRR8953103, SRR8953120, SRR8953104, SRR8953101.
Please refer to https://www.sciencedirect.com/science/article/pii/S105046482030437X, Availability of supporting data and materials
If the data has already been uploaded to NCBI, but is not yet public, then we should get this result.
Response 2: Thank you very much for your advice, we have provided the latest accession numbers (line 680-681), and the original data will be available for download on December 1, 2024.
The NCBI accession numbers: SRR28089698; SRR28089699; SRR28089700; SRR28089701; SRR28089702; SRR28089703.
Comments 3: Table 2 can be deleted, it is not necessarily to provide this data.
Response 3: Thanks for your advice, we have moved it to the supplementary material (Table S1) (line 662-663).
Comments 4: Fig 4 Please keep consistent in name of sample, for example xin, shen, nao in the Fig 4A is not matched to the HT, BR,AG in the Fig 4B.
Response 4: We couldn't agree more with you, and we have modified the name of sample in Fig 4A (line 207-210).
Comments 5: In Fig 5 , it is confused to use circle to indicated trigle, angle to up+down. Please combined the UP and DOWN data, and give the result of kegg figure. In Fig 5B, many pathway was enriched one gene, it has on meaning, delete these pathway. In Fig 5C, only one gene is enriched in KEGG, it has no meaning, delete Fig 5C.
Response 5: Thanks for your advice. These figures are only using the differentially expressed genes (DEGs) for enrichment analysis, then, we add the “up” and “down” to indicate the expression characteristics of DEGs in every pathways, but this may make it harder to understand. So, we have deleted unnecessary information in Fig 5 (line 211), and only show the enriched KEGG pathways with significant differences.
Comments 6: The expression of CRH and AVP in Fig 12, a d. And CRHBP, AVPR2, MC5R in Fig 12 b e showed reverse result, added discussion in the MS.
Response 6: Thanks for your advice, we had added the discussion in line 400-411.
Comments 7: In line 617-618 , you used long term and short term exposure noise. However, your rna-seq only used the sample of exposure of 6 days, Why. In Fig 12, you used short-term exposure fish, but you these samples are not used in rna-seq analysis.
Response 7: We only select the samples of exposure of 6 days for RNA-seq, the main reasons are as follows:
(1) previous researchers have completed short-term noise stimulation experiments on small yellow croaker, and the results showed that stimulus death did not occur [1]. This may well suggest that the yellow croakers are adaptive to short-term stimulation, but it does not answer the specific cause of death.
(2) in our study, death also occurred only in the long-term stimulation group, and occurred continuously with the duration of the stimulus. Another feature is that not all of the dead fish showed signs of cerebral hemorrhage, it suggest that there must be some adaptive regulatory mechanisms or “harbinger of death” in these surviving fishes. Thus, in order to understand the cause of death, it is more appropriate to study the fish that are still alive.
Finally, due to the limited experimental funds, we only performed RNA-seq on the samples of the long-term stimulation group, and only tested the expression level changes of HPA axis related genes in the short-term stimulation group and dead fishes of long-term stimulation group.
References:
[1] Zhang X, Zhou J, Xu W, et al. Transcriptomic and Behavioral Studies of Small Yellow Croaker (Larimichthys polyactis) in Response to Noise Exposure. Animals 2022, 12, 2061. https://doi.org/ 10.3390/ani12162061
Comments 8: As your focus is on the HPA axis, could you explain why you chose the heart as the tissue of study? What is the specific relationship between the heart and the HPA axis?
Response 8: Thank you for your kind question. In our early experiments, when we conducted experiments with different duration (10min, 30min, 1 h, 8h, 16h and 24h) of noise stimulation, we found that the time range of frequent mortality was between 16 and 24 hours, but intracerebral hemorrhages were infrequent in these dead fish. At that time we suspected cardiac resonance death caused by the underwater strong noise, and was also considered the possibility of hypertension and cardiovascular. On the other hand, through literature research, we found that the HPA axis was likely to be the key regulatory axis during noise stress, and cortisol (an important functional hormone in the HPA axis) also has a significant effect on cardiovascular and cerebrovascular diseases. Therefore, we designed the experiment with a longer stimulation duration (6 days) as described in the current manuscript, and using the brain, adrenal glands, and heart for further study.
Some references:
[1] Peña-Romo A, Gámez-Méndez A, Ríos A, et al. Noise enhanced the electrical stimulation-contractile response coupling in isolated mouse heart. International Journal of Cardiology 2016, 221, 155-160. https://doi.org/10.1016/j.ijcard.2016.06.130
[2] Schmidt F, Herzog J, Schnorbus B, et al. The impact of aircraft noise on vascular and cardiac function in relation to noise event number: a randomized trial. Cardiovascular Research 2021, 117 (5), 1382-1390. https://doi.org/10.1093/cvr/cvaa204
[3] Evans J, Torres-Pérez J, Petrazzini M, et al. Stress reactivity elicits a tissue-specific reduction in telomere length in aging zebrafish (Danio rerio). Scientific Reports 2021, 11, 339. https://doi.org/10.1038/s41598-020-79615-1
[4] Chen W, Zhen J, Tan T. Review of the regulatory mechanisms of stress responses in animals. Swine Industry Science 2021, 38(8), 35-38. https://doi.org/10.3969/j.issn.1673-5358.2021.08.010
[5] Allen J, Mezuk B, Byrd D, et al. Mechanisms of Cardiometabolic Health Outcomes and Disparities: What Characteristics of Chronic Stressors are Linked to HPA-Axis Dysregulation? Journal of Aging and Health 2022, 34(3), 448-459. https://doi.org/10.1177/08982643221085903
Comments 9: L616-618: Please explain why the author chose 10 minutes as the duration for short-term noise stress and 6 days for long-term noise stress? What was the rationale behind selecting these specific time frames for this study?
Response 9: Thank you for your kind question. In our early study, we performed 2 different noise stimulation experiments: (1) using the same noise (90-120 dB re 1 µPa), but different treatment time (10min, 30min, 1 h, 8h, 16h and 24h); (2) using different noise (60-120 dB re 1 µPa, including 60, 65, 75, 80, 90-120), but the same treatment time, including 2 min, 5 min and 10 min.
At last, we found that the death only occurred in the strong noise (90-120 dB re 1 µPa) exposure group (experiment (1)) during 16-24h, and other noise treatment group did not caused death. But intracerebral hemorrhages were infrequent in these dead fish, it was difficult to speculate on the cause of death. Finally, based on the possibility that small yellow croaker has a curve-like adaptation process to noise stimulation, we chose to set up a longer stimulation experiment for 6 days. In this way, the time of occurrence of dead fish and the differential characteristics of tissue damage can be continuously recorded and observed for a long time.
And the main reasons for choosing 10 minutes as the duration of short-term noise stress were: (1) sample collection can also cause a stress response in small yellow croaker, if noise treatment time is too short that the stress response features are likely to be covered; (2) we need take 3-5 min to finish the sample collect work of one fish. Therefore, we chose 10 min as the duration of short-term noise stress.
Reviewer 2 Report
Comments and Suggestions for Authors
This manuscript (ijms-3298388) under the title "Metabonomics and transcriptomics analyses reveal the underlying causes of underwater noise pollution causing lethality in Larimichthys polyactis by interfering HPA axis" examined the transcriptomic and metabolomic profiles of HPA axis-associated assisted tissues in Larimichthys polyactis underwater short-term and long-term noise stress. Results from this study further confirmed the damage to the peripheral and central organs in Larimichthys polyactis subjected to different noise pollutions and screened the potential regulatory pathways.
Main content of this paper is valuable for monitoring the health status in Larimichthys polyactis and other fishes exposed to the underwater noise environment. However, the current paper contains many unnecessary descriptions, particularly the main text in the section of "2. Results". Therefore, the authors need to modify the corresponding parts of this paper to improve its quality.
Major comments:
1. In the "Abstract" part, some critical information is missing, such as the treatment time of noise stress, the initial size and total number of experimental fish, and so on. Moreover, the primary results/findings of this study are required in this part. Thus, "Abstract" part should be clearly written and added more necessary information.
2. Too many unnecessary and redundant descriptions are present in the original texts of "2.Results". For example, the sentences in Line 248-249, Line 278, etc, are the methodological descriptions. They could be deleted directly without any negative influence on the corresponding paragraph or moved to the "4.Materials and Methods" part.
Additionally, many descriptions in "2.Results" are speculative and discursive, including the text of Line 202-203, 275-276, etc. It would be preferable to move them to discussion part for further comparative analysis.
Many same errors are present in the other parts of "2.Results". Thus, the main contents of the result part should be simplified and streamlined.
3. Regarding "3.Discussion" part, the current text is inadequate and contains multiple repeating interpretations of results. It is suggested to reduce the duplicate description of results. More importantly, the discussion on the conflicting/similar findings of underwater noise-induced physiological and transcriptomic changes in aquatic animals and the corresponding comparative analysis is required.
Thus, the authors should reorganize the discussion part for fully clarifying your main findings.
4.In the "4. Materials and Methods", the methodological descriptions are too complicated and detailed. Please simplify the presentation of some conventional method (such as RNA extraction, qRT-PCR, so on) or move the relevant descriptions to the supplementary part.
Moreover, it is recommended that the authors provide the ethical statement on animal experimentation and the statistical analysis in a separate part.
5. Please re-check the format of reference list carefully according to the instructions for authors. The current reference list is a bit chaotic, including wrong/missing volume and page numbers, and inconsistencies like abbreviated vs. full journal names, capitalized vs. lower-case article titles.
For example, the same journal is presented with different names in Reference 33 and 46. "Int. J. Mol. Sci." or "International Journal of Molecular Sciences", which one is correct? Similarly, the journal name is shown as the full journal name or the abbreviated journal name in this manuscript? e.g. Reference 3, 55, 57, etc. There were similar errors in the other references.
Reference 16, 23, etc, have non-italic scientific name of species in article title. The correct volume numbers and page numbers should be provided in Reference 48, 53, etc.
Thus, the authors should re-check and modify the reference list seriously.
Other errors (highlighted in yellow) were marked in the PDF file.
So, this manuscript will be reconsidered after major revision.

Author Response
Comments 1: In the "Abstract" part, some critical information is missing, such as the treatment time of noise stress, the initial size and total number of experimental fish, and so on. Moreover, the primary results/findings of this study are required in this part. Thus, "Abstract" part should be clearly written and added more necessary information.
Response 1: Thank you very much for your advice. We have added these information (Line 16-18 and 21-23), including the parameter of noise, the treatment time of noise stress, the initial size (weight and length), total number of experimental fish, the main findings and conclusions .
Comments 2: Too many unnecessary and redundant descriptions are present in the original texts of "2.Results". For example, the sentences in Line 248-249, Line 278, etc, are the methodological descriptions. They could be deleted directly without any negative influence on the corresponding paragraph or moved to the "4.Materials and Methods" part.
Response 2: We quite agree with your suggestion. We had deleted these descriptions. For details, please see the red marked section of the manuscript.
-Comments: Additionally, many descriptions in "2.Results" are speculative and discursive, including the text of Line 202-203, 275-276, etc. It would be preferable to move them to discussion part for further comparative analysis.
Response: We have moved them to the discussion part. For details, please see the red marked section of “Discussion” (line 321, 335, 417, and 455).
-Comments: Many same errors are present in the other parts of "2.Results". Thus, the main contents of the result part should be simplified and streamlined.
Response: Thanks for your advice, we have simplified and streamlined some parts of “Results” (line 101, 138, 223, 268, and 295).
Comments 3: Regarding "3.Discussion" part, the current text is inadequate and contains multiple repeating interpretations of results. It is suggested to reduce the duplicate description of results. More importantly, the discussion on the conflicting/similar findings of underwater noise-induced physiological and transcriptomic changes in aquatic animals and the corresponding comparative analysis is required.
Thus, the authors should reorganize the discussion part for fully clarifying your main findings.
Response 3: Thank you very much for your advice. We removed some repetitive content and described the discussion in sections. And we split the “Discussion” into three smaller chapters for easy reading and comprehension (line 335, 417, and 455).
Comments 4: In the "4. Materials and Methods", the methodological descriptions are too complicated and detailed. Please simplify the presentation of some conventional method (such as RNA extraction, qRT-PCR, so on) or move the relevant descriptions to the supplementary part.
Response 4: Thanks for your advice. We have simplified and deleted the non-essential information. For details, please see the red marked section of “Materials and Methods” (line 506-642).
-Comments: Moreover, it is recommended that the authors provide the ethical statement on animal experimentation and the statistical analysis in a separate part.
Response: Thanks for your advice, the ethical statement on animal experimentation have been provide in “Institutional Review Board Statement”, including the approval date and permit number. The statistical analysis of some of the experiments is also described in a simplified manner in “Materials and Methods”.
Comments 5: Please re-check the format of reference list carefully according to the instructions for authors. The current reference list is a bit chaotic, including wrong/missing volume and page numbers, and inconsistencies like abbreviated vs. full journal names, capitalized vs. lower-case article titles.
For example, the same journal is presented with different names in Reference 33 and 46. "Int. J. Mol. Sci." or "International Journal of Molecular Sciences", which one is correct? Similarly, the journal name is shown as the full journal name or the abbreviated journal name in this manuscript? e.g. Reference 3, 55, 57, etc. There were similar errors in the other references.
Reference 16, 23, etc, have non-italic scientific name of species in article title. The correct volume numbers and page numbers should be provided in Reference 48, 53, etc.
Thus, the authors should re-check and modify the reference list seriously.
Other errors (highlighted in yellow) were marked in the PDF file.
So, this manuscript will be reconsidered after major revision.
Response 5: Thank you very much for your advice. That was our negligence, we have re-checked the format of reference list, and have corrected many unnecessary errors and loopholes. For details, please see the red marked section of “References” (line 685).
Reviewer 3 Report
Comments and Suggestions for Authors
The manuscript “Metabonomics and transcriptomics analyses reveal the under-lying causes of underwater noise pollution causing lethality in Larimichthys polyactis by interfering HPA axis” investigated the effects of underwater noise stress on the small yellow croaker by transcriptomic and metabonomics analysis.
In my opinion, the topic of study could be very interesting. However, the authors must extensively revise the English of the entire manuscript. The sentences are very long and, in some cases, incomprehensible, which makes the manuscript very difficult to read. In particular, it makes it difficult to understand the methodology the authors used in their experiment and the results they obtained. I suggest that the authors ask a native speaker for support.
The introduction is very long and, although it contains important information for understanding the topic of study, it should be shorter and more direct. Authors should be careful when talking about AVP, as in the case of fish it should not be referred to as AVP but as AVT (arginine vasotocin). The methodology and results as I mentioned are difficult to understand due to the way it is written.
Finally, the bibliography should be revised as it is not uniform. There are some references that begin in capitals while others are in all lower case, and most importantly, the species names must be in italics.
Comments on the Quality of English Language
The authors must extensively revise the English of the entire manuscript.
Author Response
Comments: In my opinion, the topic of study could be very interesting. However, the authors must extensively revise the English of the entire manuscript. The sentences are very long and, in some cases, incomprehensible, which makes the manuscript very difficult to read. In particular, it makes it difficult to understand the methodology the authors used in their experiment and the results they obtained. I suggest that the authors ask a native speaker for support.
The introduction is very long and, although it contains important information for understanding the topic of study, it should be shorter and more direct. Authors should be careful when talking about AVP, as in the case of fish it should not be referred to as AVP but as AVT (arginine vasotocin). The methodology and results as I mentioned are difficult to understand due to the way it is written.
Finally, the bibliography should be revised as it is not uniform. There are some references that begin in capitals while others are in all lower case, and most importantly, the species names must be in italics.
Comments on the Quality of English Language
The authors must extensively revise the English of the entire manuscript.
Response: We really appreciate your advice and help. We have streamlined the "Introduction" (line 30), "Materials and Methods" (line 506), and "Results" sections (line 100), removed much unnecessary information and content, and changed some special terms (“AVT”, line 23, 45, 94, 221 (in table 2), 299, etc.). In the “References” (line 685), we re-checked and revised it. Meanwhile, we have conducted the English editing service of MDPI, to improve the quality of the manuscript. For details, please see the red marked section of the manuscript.
Round 2
Reviewer 1 Report
Comments and Suggestions for Authors
Please see the attached files. Thanks.

Author Response
Comments: The author only provided six SRA files. This paper included 18 SRA file (6 brain samples, 6 heart samples and 6 adrenal gland samples), and the paper will be published. Therefore, please give the evidence that you have totally uploaded 18 SRA files to NCBI.
Response: Thank you very much for your kind advice. We integrated the data packets through the NCBI system, and finally divided them into 6 groups, including 3 experimental groups (TE group of brain; TE group of heart; TE group of adrenal gland) and 3 control groups (TC group of brain; TC group of heart; TC group of adrenal gland). Each group included 3 samples, and each sample had 2 data packets. In total, there were 18 samples and 36 data packets. Please see the attachment.

Reviewer 2 Report
Comments and Suggestions for Authors
The resubmitted manuscript (ijms-3298388) under a new title "Metabonomics and transcriptomics analyses reveal the underlying HPA-axis-related mechanisms of lethality in Larimichthys polyactis exposed to underwater noise pollution" has been modified according to the specific comprehensive comments. And the responses to the corresponding comments are provided.
But some abbreviations are not defined, such as SPL, AVT, CRH, etc. It would be good to show the full name of abbreviation when first introduce it. At the first time, the full word used with the abbreviation between brackets, afterwards only the abbreviation should be used. For example, in Line 53-55.
Moreover, the whole text of the result part in the revised paper is still excessively long, containing many methodological descriptions. The statements in Line 169-172 lack of the corresponding figure or table.
In my view, the current version still needs major revision.
Author Response
Comments: But some abbreviations are not defined, such as SPL, AVT, CRH, etc. It would be good to show the full name of abbreviation when first introduce it. At the first time, the full word used with the abbreviation between brackets, afterwards only the abbreviation should be used. For example, in Line 53-55.
Response: Thanks for your advice, it's our negligence, we have modified and added this information, in Line 20, 23-24, 38, 47, 57-58, 98-99, 150, 154-155, 167-168, 286-288, 295, 298, 338-339.
Comments: Moreover, the whole text of the result part in the revised paper is still excessively long, containing many methodological descriptions. The statements in Line 169-172 lack of the corresponding figure or table. In my view, the current version still needs major revision.
Response: Thanks for your advice. We have simplified the "Results" section again, and remove the methodological descriptions (Line 106, 136, 187, 218, 245). Meanwhile, we further simplify and revise the manuscript, as detailed in the red marked section.
In addition, we modify Figure 5 according to reviewer 1's comments, delete the enrichment result of single gene in the TR_Ht_vs_Control_Ht group, that is only one gene enriched in KEGG (previous Figure 5C), it has no meaning. So we offer a new Figure 5 (Line 175) in manuscript, and add the description and explanation in Line 153-156.

Reviewer 3 Report
Comments and Suggestions for Authors
The authors have done a great job in giving more details of their work and correcting mistakes in the introduction and references, including correcting some of the English, which was the biggest problem I found. However, the English of the manuscript still needs to be revised as it still has some errors throughout the manuscript.
Comments on the Quality of English LanguageThe English of the manuscript still needs to be revised as it still has some errors throughout the manuscript.
Author Response
Comments: The authors have done a great job in giving more details of their work and correcting mistakes in the introduction and references, including correcting some of the English, which was the biggest problem I found. However, the English of the manuscript still needs to be revised as it still has some errors throughout the manuscript.
The English of the manuscript still needs to be revised as it still has some errors throughout the manuscript.
Response: Thank you very much for your kind comments and suggestions.
We have simplified and revised the manuscript again, mainly in the "Results" section (such as Line 106-116, 139-140, 145-164, 196-198, 223, 249-256, etc.) this part is also what reviewer 2 suggested us to modify.
We also have used the English Editing Service of MDPI (Edit text number: english-87363) to improve the quality (yellow marked sections in the manuscript).

Round 3
Reviewer 2 Report
Comments and Suggestions for Authors
The revised paper (ijms-3298388) entitled "Metabonomics and transcriptomics analyses reveal the underlying HPA-axis-related mechanisms of lethality in Larimichthys polyactis exposed to underwater noise pollution" has been carefully revised as suggested. The authors have given the corresponding responses/explanation in online system.
Thus, it is recommended to accept the current revision of this manuscript.